# Suspicious Positive Peritoneal Cytology (Class III) in Endometrial Cancer Does Not Affect Prognosis

**DOI:** 10.3390/jcm11216527

**Published:** 2022-11-03

**Authors:** Kenbun Sone, Eri Suzuki, Ayumi Taguchi, Harunori Honjoh, Akira Nishijima, Satoko Eguchi, Yuichiro Miyamoto, Takayuki Iriyama, Mayuyo Mori, Yutaka Osuga

**Affiliations:** Department of Obstetrics and Gynecology, Faculty of Medicine, The University of Tokyo, 7-3-1 Hongo Bunkyo-ku, Tokyo 113-8655, Japan

**Keywords:** endometrial cancer, peritoneal suspicious positive cytology, lymph node metastasis, omentectomy

## Abstract

Positive peritoneal cytology is a poor prognostic factor in patients with advanced endometrial cancer. Suspicious positive peritoneal cytology (class III) is commonly encountered in clinical practice. However, no standard treatment protocol exists for its management. Here, we investigated a possible relationship between suspicious positive peritoneal cytology, disease stage, risk factors, and endometrial cancer prognosis. We included patients diagnosed with endometrial cancer who underwent total hysterectomy and peritoneal cytology at the University of Tokyo Hospital between 2008 and 2022. Overall, 670 patients were included in the analyses; both demographic and clinical data of the patients were collected. The proportion of patients with lymph node metastasis was significantly different between peritoneal cytology groups, showing lymph node metastasis to be more extensive in patients with positive or suspicious positive peritoneal cytology than in patients with negative peritoneal cytology (*p* < 0.05). Thirty-nine patients had suspicious positive peritoneal cytology. Omental resection and biopsy were performed in 16 cases. No case of omental metastasis was found. Among patients with suspected ascites cytology, no patient experienced symptom recurrence or death. Therefore, monitoring lymph node metastasis in suspicious positive cases is essential. Moreover, a change of treatment method based on the finding of suspected positive peritoneal cytology is not necessary.

## 1. Introduction

Endometrial cancer is the most prevalent gynecological cancer [1]. Lifestyle and dietary risk factors for endometrial cancer include hypertension and diabetes mellitus. Estrogen stimulates and regulates the growth of the endometrial lining; however, low levels of luteinizing hormone and excessive estrogen levels can cause endometrial hyperplasia, which increases the risk of endometrial cancer development [1]. Other risk factors for endometrial cancer development include nulliparity, obesity, irregular menstrual periods, and estrogen hormone replacement therapy [1].

Primary treatment modalities for endometrial cancer include total hysterectomy, bilateral salpingo-oophorectomy, and lymphadenectomy [2]. Patients with a moderate-to-high risk of recurrence undergo chemotherapy and/or radiotherapy following surgery. Whilst the prognosis of early-stage endometrial cancer is good, response to treatment in advanced or recurrence cases is often refractory [2]. In the 2008 FIGO staging system, stage IIIA endometrial cancer is described as endometrial cancer with serosal or adnexal involvement. However, positive peritoneal cytology is excluded from the staging system for determining advanced stages of endometrial cancer [3].

The advantage of using positive peritoneal cytology as a prognostic factor for endometrial cancer remains unclear [4,5]. Studies have suggested positive peritoneal cytology as a prognostic factor in advanced endometrial cancer. Matsuo et al. [6] reported positive peritoneal cytology as a prognostic factor for stage II to III endometrial cancer. Moreover, 60% of patients with metastasis to the omentum had positive peritoneal cytology, which often results in the patients requiring partial omentectomy [6]. Peritoneal cytology is performed during surgery for endometrial cancer for the above reasons. Suspicious positive peritoneal cytology (class III) is often encountered in clinical practice; however, its optimal treatment method remains unclear, which complicates the choice of further treatment for clinicians. Therefore, in this study, we examined the relationship between suspicious positive peritoneal cytology and the prognosis of patients with endometrial cancer.

## 2. Materials and Methods

### 2.1. Eligibility of Patients

The study included 670 patients with endometrial cancer, who underwent total hysterectomy and peritoneal cytology at the University of Tokyo Hospital between 2008 and 2021. Patients were excluded from the study if they had uterine sarcomas, had not undergone total hysterectomy, had double cancers, and had not undergone ascites cytology.

### 2.2. Clinical Information

The following data were collected: patients’ age, histological type (endometrioid, serous, clear cell, carcinosarcoma, undifferentiated, mixed, mucinous, and others), cancer grade (1, 2, and 3), cancer stage (T1a, T1b, T2, T3a, and T3b), type of metastasis (lymph node, adnexal, and omental), and peritoneal cytology result (negative, positive, and suspicious positive). Histological type was based on the International Classification of Diseases for Oncology (third edition) or the World Health Organization histological classification schema. The main fluid cytology findings consistent with suspicious positive ascites were: (1) atypical cells with enlarged or irregular nuclei and increased levels of chromatin, and (2) tumor cells that were difficult to distinguish from other cells, such as reactive mesothelial cells, due to a low cell count. Peritoneal cytology diagnoses in this study were performed by a certified cytopathologist and cytology technologist.

### 2.3. Statistical Analyses

Statistical analyses were performed using Eazy R (EZR) statistical software, as used in a previous study [7]. Each group (negative, positive, and suspicious positive peritoneal cytology) was compared via Fisher’s exact test using a 2 × 3 contingency table. Statistical significance was set at *p* < 0.05.

## 3. Results

A total of 670 patients were included in the analysis. The median age of the patients was 55 years. Patients were categorized based on cancer stage into stage I, 483 cases (72%); stage II, 53 cases (7.9%); stage III, 117 cases (17.5%); and stage IV, 17 cases (2.5%). A total of 451 patients (68.2%) had myometrial invasion of ≤50%, and 219 patients (32.7%) had myometrial invasion of ≥50%. Lymph node metastasis was or was not found in 92 (13.7%) and 578 (86.3%) patients, respectively. Histological type comprised the following: endometrioid G1, 442 cases (66%); G2, 99 cases (14.8%); and G3, 54 cases (8%). Other histological types included serous, clear cell, and others (n = 75 cases, 11.1%). There were 24 cases (3.6%) of adnexal metastasis and 18 cases (2.7%) of distant metastasis, including omental metastasis. Peritoneal cytology was negative in 553 cases (82.5%), suspicious positive in 39 cases (5.8%), and positive in 78 cases (11.6%) (Table 1).

We examined the relationship between peritoneal cytology (negative, suspicious positive, and positive) and endometrial cancer stage (I + II vs. III + IV), risk factors, lymph node metastases, deep myometrial invasion, adnexal metastasis, distant metastasis, and pathological type. There was a significant difference in the proportion of patients with stages I + II vs. III + IV between the peritoneal cytology groups (*p* < 0.001). The proportion of patients with stage III + IV was higher in patients with positive peritoneal cytology than in patients with suspicious positive and negative peritoneal cytology (Table 2, Appendix A, Figure 1a). Additionally, the proportion of patients with lymph node metastasis (negative vs. positive vs. suspicious positive) was significantly different between the peritoneal cytology groups (*p*-value: 0.00124), showing a greater lymph node metastasis in patients with positive or suspicious positive peritoneal cytology than in patients with negative peritoneal cytology (Table 2, Appendix A, Figure 1b). Furthermore, there was a significant difference in the proportion of patients between the histological types (G1 + G2 vs. G3 + others) (*p* < 0.001). The incidence of G3 + others was higher in patients with positive peritoneal cytology than in patients with suspicious positive and negative peritoneal cytology (Table 2, Appendix A, Figure 1c).

The proportion of patients with deep myometrial invasion (>1/2 vs. <1/2) and adnexal metastasis was significantly different between the peritoneal cytology groups (Table 2, Appendix A, Figure 1d,e). Deep myometrial invasion was more extensive in patients with positive peritoneal cytology than in patients with suspicious positive and negative peritoneal cytology. Conversely, adnexal metastasis was more extensive in patients with positive and suspicious positive peritoneal cytology than in those with negative peritoneal cytology. More extensive distant metastasis was found in patients with positive peritoneal cytology than in those with suspicious positive and negative peritoneal cytology (Table 2, Appendix A, Figure 1f). None of the patients with cytological suspicions of ascites experienced recurrence or death. Among these patients, omentectomy was performed in 16 cases, although none of them had omental metastasis. (Appendix A).

## 4. Discussion

Patients with suspicious positive peritoneal cytology presented with atypical cells, which had enlarged or irregular nuclei and increased levels of chromatin. In addition, they also presented with tumor cells that were difficult to distinguish from other cells, such as reactive mesothelial cells, due to low cell counts [8]. Currently, while there are many reports of positive peritoneal cytology and endometrial cancer in the literature, few studies have explored the possibility of a relationship between endometrial cancer and suspicious positive peritoneal cytology.

Of the few existing studies, some studies reported increased rates of peritoneal cytology when several adverse prognostic factors were present. Specifically, the rate of positive peritoneal cytology was high in advanced cases, including those with lymph node, adnexal, and intraperitoneal metastases [9]. Slomovits et al. reported that positive peritoneal cytology may be a poor prognostic factor in high-risk cases (patients with deep muscle invasion, endometrioid carcinoma G3 + special histology, and positive vascular invasion), where the lesion is limited to the uterus [10]. Matsuo et al. retrospectively studied 7467 patients with endometrial cancer who underwent total hysterectomy and peritoneal cytology at stages II–III. The authors reported a significantly worse prognosis in patients with advanced endometrial cancers and positive peritoneal cytology [6]. In contrast, it was reported that positive peritoneal cytology was not a poor prognostic factor in low-risk cases, where the lesion was confined to the uterus [11,12]. However, Matsuo et al. highlighted that positive peritoneal cytology is a potential risk factor for endometrial cancer and advocate that peritoneal cytology analysis should be continued in further research [13]. The National Comprehensive Cancer Network and Japan Society of Gynecologic Oncology Guidelines recommend performing intraoperative peritoneal ascites cytology, while Matsuo et al. warn that the European Society of Medical Oncology, European Society of Gynaecological Oncology, and European Society for Radiotherapy and Oncology do not mandate performing peritoneal cytology because positive peritoneal cytology is excluded from the staging system when determining advanced stages of endometrial cancer [13,14]. In addition, several recent clinical trials for endometrial cancer included positive ascites cytology as a risk factor [13,15,16]. Thus, other investigators believe that peritoneal cytology may be an important risk factor [13,15,16], and further studies are needed to elucidate this aspect. As mentioned above, peritoneal cytology is recommended at least in association with endometrial cancer surgery in Japan, and we conducted this study because we occasionally experience peritoneal suspicious positive cases among patients with endometrial cancer.

Additionally, positive ascites cytology was reported to be significantly associated with omental metastasis, with the recommendation of omentectomy [17,18].

Although the outcomes of patients with positive peritoneal cytology have been well-studied, the surgical and postoperative treatment options of patients with suspicious positive peritoneal cytology, who are commonly encountered in clinical practice, have not been clarified. In studies that investigated suspicious positive peritoneal cytology in endometrial cancer, suspicious positive cases were grouped with positive cases, and the relationship between suspicious positive peritoneal cytology, risk factors, and prognosis was not studied in detail.

The current study showed a significant increase in the proportion of positive peritoneal cytology in patients with stage III + IV cancer than in patients with stage I + II cancer. Moreover, there was a significant increase in the proportion of positive and suspicious positive peritoneal cytology in patients with lymph node metastasis compared to that in patients without lymph node metastasis. Adnexal metastasis showed the same trend as lymph node metastasis. There was a significant increase in the proportion of positive cytology in patients with G3 or other histology compared to patients with G1 + G2 histology, whereas no increase in proportion was observed in patients with suspicious positive or negative cytology. Additionally, among cases of deep myometrial invasion and distant metastasis, the percentage of patients in the positive ascites cytology group was higher than those in the suspicious positive and negative groups.

Among patients with lymph node metastasis and adnexal metastasis, the number of patients with suspicious positive peritoneal cytology was similar to that of patients with positive peritoneal cytology. Moreover, the distribution of other risk factors in patients with suspicious positive peritoneal cytology and patients with negative peritoneal cytology was similar. This suggests that among the risk factors studied, lymph node and adnexal metastases are observed in patients with suspicious positive peritoneal cytology. These factors may correlate with cancer stage. However, only a few positive cases were found in the adnexal metastasis cases; thus, a larger sample size is required for future studies.

Of the 39 patients with suspected positive peritoneal cytology, no patient experienced recurrence or death. Among these patients, omentectomy was performed in 16 cases, but none of the patients had omental metastasis. These results suggest that peritoneal cytology is not a risk factor for recurrence, despite an increased rate of lymph node metastasis in patients with suspicious positive peritoneal cytology. As suggested in previous reports, lymph node metastasis is not a prognostic factor in cases of endometrial cancer surgery [19]. Thus, it is considered that omentectomy is not necessary for patients with positive peritoneal cytology because omental metastasis was not identified in cases of suspicious positive cytology in which omentectomy was performed.

## 5. Conclusions

The novelty of this study lies in its focus on suspicious positive peritoneal cytology in endometrial cancer. In summary, this study showed lymph node metastasis rates to be similar between patients with endometrial cancer who had suspicious positive peritoneal cytology and those who had positive peritoneal cytology. In contrast, other risk factors were similar to those of patients with negative peritoneal cytology. This suggests that it is necessary to monitor lymph node metastasis in patients with suspicious positive peritoneal cytology. In addition, omentectomy was not considered significant because there was no omental metastasis in patients with suspected peritoneal positivity. Since there were no instances of recurrence or death, it is not necessary to change the current treatment methods (chemotherapy and radiotherapy), based on the finding of suspicious positive peritoneal cytology.

## Figures and Tables

**Figure 1 jcm-11-06527-f001:**
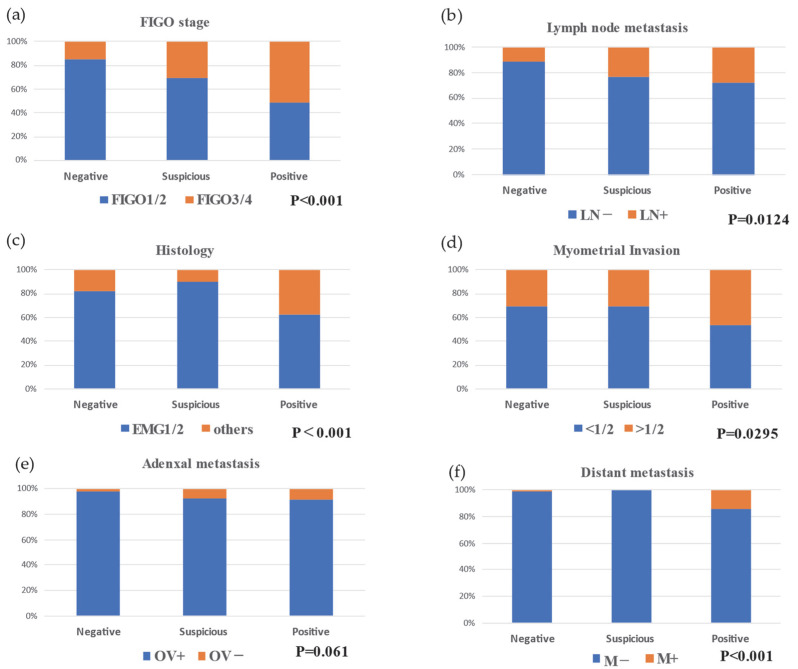
Relationship between risk factors and peritoneal cytology: (**a**) Relationship between advanced stage and peritoneal cytology; (**b**) Relationship between lymph node metastasis and peritoneal cytology; (**c**) Relationship between histology and peritoneal cytology; (**d**) Relationship between myometrial invasion and peritoneal cytology; and (**e**) Relationship between adnexal metastasis and peritoneal cytology. FIGO, the International Federation of Gynecology and Obstetrics; EMG1/2/3, endometrioid cancer grade 1/2/3. (**f**) Relationship between distant metastasis and peritoneal cytology.

**Table 1 jcm-11-06527-t001:** Patient characteristics.

	All Patients (n = 670)
Age [year], median (range)	55 (23–93)
FIGO stage, n (%)	
I	483 (72%)
II	53 (7.9%)
III	117 (17.5%)
IV	17 (2.5%)
Myometrial invasion, n (%)	
<1/2	451 (67.3%)
≥1/2	219 (32.7%)
Lymph node metastasis, n (%)	
Negative	578 (86.3%)
Positive	92 (13.7%)
Histology, n (%)	
EMG1	442 (66%)
EMG2	99 (14.8%)
EMG3	54 (8.0%)
Others	75 (11.1%)
Distant metastasis, n (%)	18 (2.7%)
Adnexal metastasis, n (%)	24 (3.6%)
Peritoneal cytology, n (%)	
Negative	553 (82.5%)
Suspicious	39 (5.8%)
Positive	78 (11.6%)

FIGO, the International Federation of Gynecology and Obstetrics; EMG1/2/3, endometrioid cancer grade 1/2/3.

**Table 2 jcm-11-06527-t002:** Relationship between risk factors and peritoneal cytology.

	Negative (n = 553)	Suspicious (n = 39)	Positive (n = 78)
Age, median (range)	56 (23–93)	53 (33–78)	54.5 (26–79)
FIGO stage, n (%)			
I	424 (76.7)	25 (64.1)	34 (43.6)
II	47 (8.5)	2 (5.1)	4 (5.1)
III	75 (13.6)	12 (30.8)	30 (38.5)
IV	7 (1.3)	0 (0)	10 (12.8)
Myometrial invasion, n (%)			
<1/2	382 (69.1)	27 (69.2)	42 (53.8)
≥1/2	171 (30.9)	12 (30.8)	36 (46.2)
Lymph node metastasis, n (%)			
Negative	61 (11.0)	9 (23.1)	22 (28.2)
Positive	492 (89.0)	30 (76.9)	56 (71.8)
Histology, n (%)			
EMG1	384 (69.4)	26 (66.7)	32 (41.0)
EMG2	73 (13.2)	9 (23.1)	17 (21.8)
EMG3	42 (7.6)	4 (10.3)	8 (10.3)
Others	54 (9.8)	0 (0)	21 (26.9)
Distant metastasis, n (%)			
Negative	7 (1.3)	0 (0)	11 (14.1)
Positive	546 (98.7)	39 (100)	67 (85.9)
Ovarian metastasis, n (%)			
Negative	14 (2.5)	3 (7.7)	7 (9.0)
Positive	539 (97.5)	36 (92.3)	71 (91.0)

FIGO, the International Federation of Gynecology and Obstetrics; EMG1/2/3, endometrioid cancer grade 1/2/3.

## Data Availability

All data generated or analyzed during this study are included in this published article.

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
