# Peer review of "Suspicious Positive Peritoneal Cytology (Class III) in Endometrial Cancer Does Not Affect Prognosis"

_jcm, 2022, doi:10.3390/jcm11216527_

Round 1

Reviewer 1 Report

The work was performed according to high methodological standards, and on an impressive number of patients.

However, in my opinion, there are few minor areas of weakness.

The term "suspicious positive" peritoneal cytology is not completely explained. Was an additional immunostaining performed during the clarification process?

Out of 670 patients, only 39 fell into the "suspicious" category. Since this diagnosis can be quite subjective, the authors should consider double checking by two cytopathologists, especially since the main result is based on this category.

Most of the results are quite obvious, logical, and expected: "a significant increase in the proportion of positive peritoneal cytology in patients with stage III + IV cancer than in patients with stage I + II cancer " or "a significant increase in the proportion of positive and suspicious peritoneal cytology in patients with lymph node metastases than in patients without lymph node metastases" (lines from 166 to 176).

Author Response

  1. The term "suspicious positive" peritoneal cytology is not completely explained. Was an additional immunostaining performed during the clarification process?

Response: Thank you for pointing this out. No additional immunostaining was performed. We diagnosed patients with suspicous positive peritoneal cytology based on the following general findings: (1) Atypical cells with enlarged or irregular nuclei and increased levels of chromatin and (2) Tumor cells that are difficult to distinguish from other cells such as reactive mesothelial cells due to a low cell count. Diagnosis of peritoneal cytology in this study was performed by a certified cytopathologist and a cytology technologist The use of general findings, without immunostaining results, allow the incorporation of the analysis in daily clinical practice, which we believe is a strength of the study. We have revised the manuscript according to your comments (Page 2, Lines 68-71)

  1. Out of 670 patients, only 39 fell into the "suspicious" category. Since this diagnosis can be quite subjective, the authors should consider double checking by two cytopathologists, especially since the main result is based on this category.

Response: Thank you for pointing this out. In this study, diagnosis of peritoneal cytology was performed by a certified cytopathologist and a technologist. Therefore, we believe the diagnosis is objective. We have revised the manuscript according to your comments (Page 2, Lines 71-72)

Most of the results are quite obvious, logical, and expected: "a significant increase in the proportion of positive peritoneal cytology in patients with stage III + IV cancer than in patients with stage I + II cancer " or "a significant increase in the proportion of positive and suspicious peritoneal cytology in patients with lymph node metastases than in patients without lymph node metastases" (lines from 166 to 176).

Response: Thank you for highlighting this. The novelty of our study lies in its focus on suspicious positives peritoneal cytology, not positive peritoneal cytology, in endometrial cancer.

As you pointed out, the findings are to be expected in patients with positive peritoneal cytology, but we believe our results are novel in terms of suspicious positive cases. We also consider our finding of an increase in lymph node metastasis in cases of suspicious positive ascites cytology to be novel.

Reviewer 2 Report

Thank you for allowing me to review this manuscript.

The paper analyzed the role of positive peritoneal cytology in EC. 

The main limitation of the present study included the lack of novelty. 

Other authors focused on this features (PMID: 32962894)

The authors have to address in the text that already the FIGO staging system (PMID: 34274133), the ESGO recommendation (PMID: 33397713) and the Rare Tumor Working Group guidelines for uterine serous carcinoma (PMID: 33934848), and clear cell endometrial cancer (PMID: 35063279) evaluated this issue.

Please improve your discussion, evaluating in deep the role of positive cytology in the light of the currently available recommendations 

Author Response

Thank you for allowing me to review this manuscript.

The paper analyzed the role of positive peritoneal cytology in EC.

The main limitation of the present study included the lack of novelty.

Other authors focused on this features (PMID: 32962894)

The authors have to address in the text that already the FIGO staging system (PMID: 34274133), the ESGO recommendation (PMID: 33397713) and the Rare Tumor Working Group guidelines for uterine serous carcinoma (PMID: 33934848), and clear cell endometrial cancer (PMID: 35063279) evaluated this issue.

Please improve your discussion, evaluating in deep the role of positive cytology in the light of the currently available recommendations

Response:

Thank you for indicating this. The novelty of our study is not about positive peritoneal cytology, but about suspicious positive peritoneal cytology in endometrial cancer. We have revised our paper to clarify the novelty of our paper (Page 6, Lines 200-201).

There are many papers on positive peritoneal cytology in endometrial cancer, as you mentioned, but there are very few papers on suspicious positive peritoneal cytology. Actually, a PubMed search did not reveal any report of suspicious positive peritoneal cytology alone in endometrial cancer. Therefore, we believe this paper will contribute to the treatment of endometrial cancer.        

Round 2

Reviewer 2 Report

The authors did not address the comments of my previous revision.

Author Response

Response to the Reviewer’s Comments:

Reviewer: 2

The authors did not address the comments of my previous revision.

Response:

We are very sorry that we misunderstood your comment. We have now revised our paper, evaluating in depth the role of positive cytology in light of the currently available recommendations, and we have also cited the paper you referred to(Page 6, Lines 160–175). However, as we have previously explained, the novelty of our study is not in relation to the positive peritoneal cytology, but instead about suspicious positive peritoneal cytology in endometrial cancer. We have revised our paper to clarify the novelty of our paper, as follows (Page 7, Lines 215–216):“The novelty of this study lies in its focus on suspicious positive peritoneal cytology in endometrial cancer.”